# Characterization and Application in Packaging Grease of Gelatin–Sodium Alginate Edible Films Cross-Linked by Pullulan

**DOI:** 10.3390/polym14153199

**Published:** 2022-08-05

**Authors:** Shuo Li, Min Fan, Shanggui Deng, Ningping Tao

**Affiliations:** 1College of Food Science and Technology, Shanghai Ocean University, Shanghai 201306, China; 2Shanghai Engineering Research Center of Aquatic-Product Processing & Preservation, Shanghai 201306, China; 3College of Food and Pharmacy, Zhejiang Ocean University, Zhoushan 316000, China

**Keywords:** pullulan, gelatin, sodium alginate, oil oxidation, edible film, grease packaging

## Abstract

Gelatin–sodium alginate-based edible films cross-linked with pullulan were prepared using the solution casting method. FTIR spectroscopy demonstrated the existence of hydrogen bonding interactions between the components, and scanning electron microscopy observed the component of the films, revealing electrostatic interactions and thus explaining the differences in the properties of the blend films. The best mechanical properties and oxygen barrier occurred at a 1:1 percentage of pullulan to gelatin (GP11) with sodium alginate dosing for modification. Furthermore, GP11 demonstrated the best thermodynamic properties by DSC analysis, the highest UV barrier (94.13%) and the best oxidation resistance in DPPH tests. The results of storage experiments using modified edible films encapsulated in fresh fish liver oil showed that GP11 retarded grease oxidation by inhibiting the rise in peroxide and anisidine values, while inappropriate amounts of pullulan had a pro-oxidative effect on grease. The correlation between oil oxidation and material properties was investigated, and water solubility and apparent color characteristics were also assessed.

## 1. Introduction

The oxidative stability of oily foods, especially those containing high levels of unsaturated fatty acids, is of significant concern. Traditional stabilization technologies (including the removal of oxygen and catalysts) and addition of antioxidants should be further studied for their safety, synergistic effect due to the combination of different components, as well as the use of packaging material [1]. Furthermore, food packaging systems’ primary function is to separate food from the surrounding environment, reducing interaction with spoilage factors (such as microorganisms, water vapor, oxygen and off-flavors) and avoiding losses of desirable compounds [2]. The factors or mechanisms that influence the oxidation of oils and fats are not only oxygen but also temperature [3], light (photosensitive oxidation), ions [4], enzymes (enzymatic oxidation), etc. [5]. Studies on packaging to retard oxidation of fats and oils have focused on foods with high fat content, such as meat [6,7] or systems such as emulsions and microencapsulation [8,9], while no experiments have been reported on packaging materials in direct contact with pure oils in bags.

In China, there is a large market for convenience foods, such as instant noodles or some vermicelli, soups, etc., accompanied by small packages of seasonings or flavored oils. Usually, the material used to seal this type of food is mainly plastic. It is inconvenient to squeeze clean, polluting the environment and causing waste during its usage. This is because these materials are non-biodegradable and non-renewable [10]; less than 3% of waste plastics are recycled globally [11]. If the packaging material of this type of food is edible, it will solve the above problems and make small bags of seasoning with convenience similar to “laundry gel”. The material needs to have good hydrophilic and oleophobic properties and meet the strength requirements in transportation to achieve this. Currently, most of the reported raw materials for developing edible packaging materials are edible polymers, mainly composed of polysaccharides, proteins and lipids, which can be easily consumed by animals and humans without causing harm to the body [12]. In addition, improving the mechanical properties, light, water and gas permeability and functionality of the materials are the primary motivations for developing polysaccharide, protein and lipid composites. These materials can be designed to extend the shelf life of food products [13,14]. Edible films are mainly formed using the gelation properties of edible polymers, often held together by electrostatic, hydrogen bonding, hydrophobic, van der Waals forces or a combination thereof, such as gels prepared from botanical hydrocolloids [15] polysaccharides, protein or making starch-based films [16,17,18].

As a food-grade material, gelatin is widely sourced, it is made from partial hydrolysis of collagen, it has good UV absorption and it is often used as the primary material for edible packagings [19], such as capsule shells and microcapsule wall materials [1]. Films made from pure gelatin are transparent, odorless and have low oxygen permeability, but are fragile such as glass, thus often need to be modified to obtain functionality [20,21], mainly in mechanical properties or antimicrobial applications [22,23], and they are mostly applied as polysaccharide-protein structures such as using cellulose. [24]. Sodium alginate is an anionic marine polysaccharide containing β-(1→4)-linked D-mannuronic acid and α-(1→4)-linked L-gulonate residues [25]. It has good gelation properties. Alginate can form dimers with divalent cations and subsequently form weak inter-dimeric aggregates governed by electrostatic interactions, then alginate will be negatively charged in solution [26]. Thus, people could apply electrostatic interactions to cross-link it with other components.

Pullulan is a microbial polysaccharide consisting of α-(1,6) glycosidic units interlinked with maltotriose units, synthesized from the starch of the sprouting stunt mold [27]. The oleophobic nature of the several hydroxyl groups contained in the structure is the property that we aim to exploit. Pullulan is highly water-soluble and capable of developing edible films. These films are colorless, odorless, heat sealable, water permeable, transparent and have low oxygen permeability. It has been reported that people have used it to keep some foods fresh, for example, developing edible coating for strawberry preservation, edible film for brussels sprouts, etc. [28,29].

Based on the above, this study aims to investigate the interaction of pullulan with gelatin and edible film components, investigate the effect of pullulan on the physicochemical properties of edible gelatin films modified by sodium alginate and provide an option for producing edible seasoning packets. Pullulan itself is not charged, so it would be subjected to charged polymers, and the weakly alkaline environment provided by sodium tripolyphosphate would make gelatin negatively charged; thus, the components would self-assemble and form a film due to electrostatic interaction. FTIR and microstructural observations were used to investigate the interaction of components. Mechanical properties were measured by tensile strength and elongation at break. Oxygen permeability and light barrier were also measured due to the need to prevent the oxidation of oils. Water solubility was measured to meet the film’s ready-to-use characteristics. DSC was used to analyze thermal stability and color characteristics, and oxidation resistance was also measured. 

## 2. Materials and Methods

### 2.1. Materials and Reagents

Pullulan (C_37_H_62_O_30_)_n_, (M_W_ ≈ 2.5 × 10^5^ Da) was obtained from Macklin Biochemical Co., Ltd. (Shanghai, China). Gelatin was developed from bovine skin, provided by Solarbio Science & Technology Co., Ltd. (Beijing, China). Sodium alginate (C_6_H_7_NaO_6_)_n_, F.W. (198.11)_n_, viscosity (10 g/L, 20 °C) ≥ 0.02, containing 30.0–35.0% ash, was obtained from Sinopharm Chemical Group (Shanghai, China). Sodium tripolyphosphate, Mw = 367.86, and food-grade glycerol were purchased from Macklin Biochemical Co., Ltd. (Shanghai, China). Fish liver was provided by Zhongyang Ecological Fish Co., Ltd. (Jiangsu, China). Soybean oil was purchased from Yihai Jiali Grain & Oil Food Co., Ltd. (Shanghai, China). 2, 2-Diphenyl 1-Picrylhydrazyl (DPPH), P-anisidine, 1, 1, 3, 3-tetra ethoxy propane (purity ≥ 97%), sodium thiosulfate standard solution, trichloroacetic acid, trichloromethane and other reagents were of analytical grade and purchased from Macklin Biochemical Co., Ltd. (Shanghai, China).

### 2.2. Preparation of Edible Films

The edible films were prepared by the solution casting method [29,30,31]. Briefly, gelatin, pullulan, sodium alginate and sodium tripolyphosphate were dissolved in distilled water at 50–60 °C, respectively, where sodium tripolyphosphate can be used as a moisture-retaining agent [32,33,34] and also as a cross-linking agent to enhance protein cross-linking [35,36,37,38]. Glycerol was added as a plasticizer. Sodium alginate, sodium tripolyphosphate and glycerol were used in equal amounts in each group, and the gelatin mixed with pullulan solutions were prepared in the proportions of 0:1, 1:3, 1:1, 3:1 and 1:0 (*w*/*w*), named GP01, GP13, GP11, GP31 and GP10. Pure gelatin (GEL) and pure pullulan (PUL) membranes were prepared for comparison. The specific formulation for each group is shown in Table 1 (on a dry weight basis). All components were mixed in proportion to develop a film-forming solution, and the film-forming solution was sonicated and degassed for 15 min to remove air bubbles, then pipetted onto clean and dry Petri dishes (Φ = 150 mm), left at room temperature for three days to stabilize and tested within three days. Each was made from 25 mL of film-forming solution per film.

### 2.3. Characterization

#### 2.3.1. Thickness

Film thickness was measured using a digital spiral micrometer (Aladdin Bio-Chem Technology Co., Ltd, Shanghai, China) with 0.001 mm. The average thickness (taken from five random locations on each sample) was used to determine the film’s mechanical properties. 

#### 2.3.2. Mechanical Property

The tensile strength (TS) and elongation at break (EAB) of the films were tested with an intelligent electronic tensile tester (XLW (EC), Jinan, China) at 25 °C and 90% relative humidity (RH). The samples were cut into 110 × 15 mm strips with an initial clamping distance of 50 mm and stretched at a speed of 50 mm/min. The TS and EAB were calculated by the following equations:(1)$$TS\;(MPa)\;=\;\frac{F}{h\;\times \;d},$$
(2)$$EAB\;\left(\%\right)\;=\;\frac{{L\;-\;L}_{0}}{L_{0}}\;\times \;100\%,$$
where *F* is the maximum force (N), *h* is the thickness (mm), *d* is the width of the membrane (mm), *L* is the length of the membrane when it breaks (mm) and *L*_0_ is the initial length of the membrane (mm) [39]. Each film sample was measured five times.

#### 2.3.3. Oxygen Transmission Rate (OTR)

The OTR of the films was measured by a gas permeability tester (Labthink, G2/132, Jinan, China). Film samples were prepared with an area of Φ = 97 mm, and the test chamber was set at 23 ± 0.1 °C with 50% ± 1% RH. Each film was determined by the differential oxygen pressure method for 12h. Each film sample was measured five times. The results were presented as OTR (cm^3^/(m^2^⋅24 h⋅0.1 MPa)).

#### 2.3.4. Water Solubility (WS)

The WS was measured using the method of Zhou et al. [39] with some modifications. Briefly, the samples were cut into film sheets (2.0 × 2.0 cm) and placed in 10 mL of deionized water, stirred with a magnetic stirrer until completely dissolved and the dissolution time was recorded. The solubility of the water was calculated from the following equation:(3)$$WS\;=\;\frac{m}{t},$$
where *m* is the mass of the membrane sheet (g), and *t* is the time taken for it to dissolve (s). Each film sample was measured five times.

#### 2.3.5. SEM

The microscopic morphology of the films was observed with a scanning electron microscope (Hitachi S3400, Tokyo, Japan). Samples were embrittled with liquid nitrogen and then placed in ion sputtering and gold sprayed at 15 mA for 60 s to improve their conductivity. The cross-sectional structure of the different modified films and the compatibility of the components were observed at an accelerating voltage of 5 kV.

#### 2.3.6. Fourier Transform Infrared (FTIR) Spectroscopy Analysis

The chemical composition of the films was analyzed using FTIR spectroscopy (Thermo Nicolet Corporation, Waltham, MA, USA). Each film was scanned 64 times from 4000 to 500 cm^−1^ with a scanning interval of 4 cm^−1^ and the air spectrum was used as a background correction.

#### 2.3.7. Thermal Stability

Differential scanning calorimetry (DSC) tests were carried out using a DSC furnace (Q2000, TA instruments, New Castle, DE, USA). Film samples (5–10 mg) were placed in aluminum pans and heated from 30 °C to 220 °C. The ramp rate was 10 °C/min, and the nitrogen flow rate was 50 mL/min. 

#### 2.3.8. Transparency and Color

The transparency of the films was studied in the wavelength range of 365–940 nm using a solar film transmission meter (LS101, Lin Shang Technology Company, Guangdong, China). The visible light transmission rate (VLT, 380–760 nm), ultraviolet light rejection rate (UVR, 365–380 nm) and infrared light rejection rate (IRR, 760–940 nm) were analyzed using air as a blank. Each film sample was measured five times.

The color parameters (L*, a*, b*) of the films were measured with a colorimeter. (Konica Minolta, CR-400, Tokyo, Japan). L* values indicate black/white (0/100), a* values indicate green/red (−80/100) and b* values indicate blue/yellow (−80/70). According to Chen et al. [40], The film sample was placed on top of a white standard plate (L* = 94.69, a* = −1.98, b* = −0.93) for testing. Each film sample was measured five times. The total color difference (ΔE) was calculated according to the following equation:(4)$$\Delta E\;=\;\sqrt{{\left(\Delta L^{*}\right)}^{2}\;+\;(\Delta a^{*}{)}^{2}\;+\;(\Delta b^{*}{)}^{2}.}$$

#### 2.3.9. DPPH Antioxidant Analysis

The DPPH antioxidant analysis was determined according to the method modified by Jiang et al. [41]. Five pieces (25 mg each) of sample film were cut and dissolved in a test tube containing 5 mL of 75% ethanol and then 5 mL of DPPH solution was added. The reaction solution was sealed and protected from light at 60 °C for 2 h. The DPPH radical scavenging ability of the membrane solution was measured at 517 nm. The formula can be expressed as follows:(5)$$DPPH\;=\;\left(1\;-\;\frac{A_{i\;}{-\;A}_{j}}{A_{0}}\right)\;\times \;100\%,$$
where *A_i_* is the absorbance of the DPPH solution mixed with the membrane solution, *A_j_* is the absorbance of the membrane solution and *A*_0_ is the absorbance of the DPPH solution.

### 2.4. Application of Films in Fish Liver Oil Storage

#### 2.4.1. Preparation and Bagging of Fish Liver Oil

To avoid the antioxidant effect of food additives in the oil products available on the market [9], we used fish liver with oil content (35.23 ± 1.76%) to develop self-made liver oil, which was prepared as follows: 10% soybean oil and 90% fish liver were weighed and fried at 140 °C for 2 min, then the liver oil was collected. The initial peroxide value (PV), anisidine value (An.V), malondialdehyde (MDA) content and total oxidation value (TOV) of the produced liver oil were measured.

The determination of the antioxidant properties of liver oil was carried out in our laboratory in advance, using a Racimat lipid oxidation stability analyzer (Aptar, Switzerland) to determine its oxidation endpoints. The liver oil oxidation curves were obtained by the Q10 extrapolation method [42] as follows:(6)$${t\;=\;4313\;\times \;e}^{-0.06968\;\times \;T}\;R^{2}\;=\;0.99883,$$
where *T* is the temperature (°C), and *t* is the storable time (h).

The resulting film materials were cut and heat-sealed using two pieces to develop bags of 8 × 8 cm, filled with 10 g of liver oil, sealed and accelerated in an oven at 40 °C. The experimental temperature was substituted into Equation (6), and we obtained *t* = 265.65 h. All bags were stored for 12 days, with oil with no encapsulation as control group 1 (blank) and PVC plastic sealed bags as control group 2 (control), and samples were taken every 3 days for testing. The GP01 film was too self-absorbent to meet the bag-making conditions and was not subjected to encapsulation.

#### 2.4.2. Determination of Malondialdehyde (MDA) in Liver Oil

The determination of MDA in oil was conducted according to the China National Food Safety Standard GB5009.181-2016. The absorbance at 532 nm was measured and quantified by comparison with a standard series curve using the following formula [43]:(7)$$X\;=\;\frac{c\;\times \;V}{m},$$
where *X* is the amount of MDA in the sample (mg/kg); *c* is the concentration of MDA in the sample solution obtained from the standard series curve (μg/mL); *V* is the volume of the sample solution (mL); *m* is the mass of the sample represented by the final sample solution (g). The initial MDA content of the liver oil was 1.25 ± 0.09 mg/kg.

#### 2.4.3. Determination of Peroxide Value (PV), Anisidine Value (An.V) and Total Oxidation Value (TOV) in Cod Liver Oil

The PV was determined by titration with sodium thiosulphate standard solution, reference to the Chinese National Food Safety Standard GB 5009.227-2016, and the results were presented in mmol/kg. The initial PV of the liver oil produced was 6.6 ± 0.35 mmol/kg.

The An.V of liver oil was measured in dimensionless units according to the International Food Safety Standard ISO6885: 2006 and the Chinese National Food Safety Standard GB/T 24304-2009: Animal and vegetable fats and oils—Determination of anisidine value. 

The total oxidation value was calculated by the following formula [44], which helps to evaluate the oxidative deterioration of the oil.
(8)TOV = PV + 4An.V.

The initial *An.V* of the resulting liver oil was 2.31 ± 0.62. The initial *TOV* was 28.72 ± 2.00.

### 2.5. Statistical Analysis

Experimental data during storage were obtained from independent triplicates. The data were compared using one-way analysis of variance (ANOVA) using SPSS 26 software (SPSS Institute Inc., Chicago, IL, USA), and *p* < 0.05 was considered to have significant differences between the data. The data were expressed as mean ± standard deviation (SD). OriginPro 2021 (Origin Lab, Northampton, MA, USA) was used to process and generate images.

## 3. Results and Discussion

### 3.1. FTIR Analysis

Figure 1 show the FTIR spectra of the pure gelatin film (GEL), the pure pullulan film (PUL) and the composite films with different gelatin–pullulan ratios. The specific functional group shared by GEL and PUL was a broad peak at 3283 cm^−1^ (O-H stretching vibration). The typical functional group of the GEL was 1636 cm^−1^ (C=O stretching of amide-I) and 1538 cm^−1^ (N-H stretching of amide-II). While the typical functional group of the GEL was at 1108 cm^−1^ (C-O stretching vibration of the primary alcohol), the strong peak at 1010 cm^−1^ of the PUL was caused by the C-O-C stretching of the (1–4) glycosidic bond linkage [22,45]. 

For PUL and GEL membranes, the peaks around 3300 cm^−1^ and 2937 cm^−1^ are related to the stretching vibration of –OH and the C-H contraction vibration in the methyl group, respectively [27,31]. The intensity of the peaks can reflect the strength of intermolecular hydrogen bonding. It can be seen that the intensity of the broad peak at 3283 cm^−1^ of GP10 co-blended membrane was not high, indicating that the intermolecular hydrogen bonding was not strong, while the intensity of this peak increased after the addition of pullulan, indicating that the intermolecular hydrogen bonding was enhanced [39], Han et al. [27] found that the conjugation of pullulan with egg white protein could increase the absorption of the hydroxyl stretching band (3300 cm^−1^).

Secondly, changes in the peaks of the blend films at 1636 cm^−1^ and 1538 cm^−1^ were also evident, indicating C=O stretching in the amide-I band, bending and stretching of N-H bonds or C-N stretching in the amide-II band, respectively [6]. The curves of the GP01 showed that C=O bonds could be formed between pullulan and sodium alginate even without the addition of gelatin. In the multiple gelatin structures, the pullulan was able to ensure the structural stability of the amide I and II bands, such as GP11 and GP31. Meanwhile, the effect of pullulan on the amide II band was more evident in the multiple PUL structures, indicating that the microstructure of gelatin was changed. This was demonstrated by subsequent experiments. In turn, the gelatin in the blended films had a similar effect on the characteristic peaks of the pullulan. For example, the change in peak intensity around 1010 cm^−1^ indicating the C-O-C bond, suggested that the structure of the pullulan had also been changed. In addition to this, GP10, GP11 and GP13 also showed new peaks around 887 cm^−1^, which may be due to the deformation vibration of O-H, while the formation of hydrogen bonds in O-H may be due to the Maillard reaction or the glycosylation of gelatin proteins during the film formation process [27], and the polysaccharide reacting with gelatin in GP10 may be sodium alginate.

### 3.2. Microstructure

Figure 2 show the cross-sectional microstructure of different films. As can be seen from the SEM images, the smooth cross-section of the GP01 indicated that the components were compatible, which was due to a large number of hydrogen bonds between the pullulan and sodium alginate. The GP10 showed a fibrous structure of gelatin protein; meanwhile, the incompatible particles were sodium alginate molecules, mainly on the same side, indicating that the sodium alginate molecules were repelled by electrostatic forces. With the addition of pullulan, the sodium alginate molecules appeared on both sides, indicating that the sodium alginate molecules were still subject to electrostatic repulsion. Due to the co-mixing of the solution, the pullulan and gelatin were bonded to each other through hydrogen bonding and van der Waals forces [16]. However, by observing the GP13 and GP31, we could find that they showed more incompatible granular structures. Our explanation for this is that both the multiple gelatin and pullulan structures could disrupt the electrostatic equilibrium between the charged gelatin molecules and the sodium alginate molecules, resulting in hydrogen bonding being weaker or stronger than the charge interactions. Specifically, the GP13 was a multiple pullulan structure whose hydrogen bonding interaction with gelatin was greater than the electrostatic repulsion of gelatin by sodium alginate leading to agglomeration of gelatin molecules into globular protein rolling, thus reducing compatibility [24]. Comparatively, the higher content of gelatin in the GP31 had a greater effect on the electrostatic attraction and hydrogen bonding of pullulan than that of sodium alginate, resulting in the agglomeration of pullulan molecules and reduced compatibility. In this regard, Figure 3 explain the reasons for this phenomenon.

### 3.3. Thickness

The thicknesses of the films developed in each ratio are shown in Table 2. The thicknesses varied when the solutions were poured in the same volume and with the same bottom area, probably due to the agglomeration of pullulan forming a non-uniform structure of the gelatin–sodium alginate film matrix in a non-continuous phase [28] and due to the different amounts of gelatin and the different degrees of electrostatic interaction with the anionic sodium alginate molecules, resulting in their different thicknesses. The GP11 film, for example, was the thickest at 0.201 mm, indicating that the parallel bonding of the polysaccharide molecular chain segments to the fibrin of the gelatin was good and that the uniform interaction gave it a thicker thickness, which had a corresponding effect on its permeability.

### 3.4. Oxygen Permeability

Table 2 also show the oxygen permeability of the different films. The determination of the oxygen permeability of the films is of great importance as it is an investigation into the role of preventing the oxidation of grease [1,3]. Of these, the GP01 was not tested this time because it was almost impossible to unfold due to its strong self-adhesive properties. We speculated that the reason for this phenomenon was that the GP01 had a large number of hydrogen bonds in the ratios of pullulan and sodium alginate. Furthermore, glycerol as a plasticizer also had hydrogen bonds, and there was no gelatin to bind the hydrogen bonds; hence the phenomenon of self-adhesion appeared after film formation. From the OTR, we can see that the PUL was a more substantial oxygen barrier than the GEL, which might prove that the pullulan film had a more dense structure than the gelatin film. The oxygen permeability of the GP10 modified with sodium alginate increased by 60.87% compared to the gelatin film, which did not favor the protection of oils and fats. The addition of pullulan increased the oxygen barrier of GP11, GP31 and GP13 by 17.80%, 5.58% and 14.96%, respectively. We found that although pullulan has a dense structure, the higher content of pullulan did not have a better oxygen barrier. The difference between the GP11 and GP13 was not significant (*p* > 0.05). This may be because the unsuitable ratio of gelatin to pullulan led to uneven interaction forces, causing agglomeration of polysaccharide molecules, which affected the compatibility, thus creating gaps in the membrane material and enhancing oxygen permeability [18].

### 3.5. Solubility

As can be seen from the solubility of the films with different ratios in Table 2, there was no significant difference between most of the films (*p* > 0.05), indicating that the use of sodium alginate, pullulan cross-linked with gelatin did not affect the sensitivity of the material to water. On the one hand, this was because the various raw materials are themselves hydrophilic edible polymers, precisely because the polar part of the membrane material (e.g., hydroxyl functional groups) hydrates with water molecules and forms hydrogen bonds. On the other hand, the membrane material itself also has free hydrogen bonds, which could offer the possibility of subsequent use of such membrane materials in combination with active substances to enrich their functionality. The water solubility of gelatin or pullulan is also a significant requirement for producing packaged edible pouches.

### 3.6. Mechanical Properties

Figure 4 assess the mechanical properties of the films with different ratios which correlates with the structural integrity, strength and flexibility of the film, including tensile strength (TS, MPa) and elongation at break (EAB, %). Both PUL and GEL exhibited high TS and low EAB, consistent with their inherent brittle nature [19,29]. The GP10 modified film without the addition of pullulan showed a significant decrease in TS (*p* < 0.05) but an increase in EAB to 45%, which was attributed to the modification of gelatin by sodium alginate and glycerol, presumably due to the homo-charge repulsion of the anion-aggregated alginate and negatively charged gelatin molecules resulting in a decrease in strength, while the cross-linked structure of gelatin was the main reason for maintaining its ductility. With the addition of pullulan, the TS of GP11, GP13 and GP31 films all improved, and the groups with more pullulan had higher EAB (GP11, GP13), indicating that pullulan could improve the mechanical properties of gelatin-based films, the multiple pullulan structures had higher EAB, relatively, the multiple gelatin structures had stronger TS, the improvement in mechanical properties was related to the reduction of particles formed by agglomeration of polysaccharide, the formation of hydrogen bonds and the formation of new bonds between polysaccharides and proteins through the Maillard reaction [27]. Interestingly, there was no significant difference in mechanical properties between the GP01 and the GP10, suggesting that the hydrogen bonding enriched by pullulan and sodium alginate in the formulation of the GP01 was the main reason for maintaining the mechanical properties of the film.

### 3.7. Thermodynamic Properties

The thermal denaturation temperature (*T_d_*), the melting temperature (*T_m_*), the latent heat of thermal denaturation (Δ*H_d_*) and the latent heat of melting (Δ*H_m_*) of different film materials were determined by DSC analysis. The DSC heat flow diagrams of GEL, PUL and composite films with different gelatin–pullulan ratios are shown in Figure 5. 

As can be seen from Figure 5, for the GEL and PUL, the first small heat absorption peak was in the evaporated water binding in the membrane, indicating the start of melting, with a *T_d_* of 138.01 °C and 148.14 °C, respectively, while the GP01 had a very high *T_d_* value, reaching 175.66 °C. The increase in Δ*H_d_* indicated a large amount of hydrogen bonding in the membrane molecules. The thermodynamic properties of the GP11 were very similar to those of the GP01, indicating that the GP11 film’s molecules also had strong hydrogen bonding. The GP10, GP31 and GP13 showed relatively similar thermodynamic properties. They did not exhibit significant heat absorption peaks, which was considered due to the high content of bound water in the films [46]. The Δ*H_d_* of the GP01, GP10, GP31, GP13 and GP11 modified films were 2.33, 1.48, 2.25, 2.10 and 1.47 times higher than that of the pullulan film and 3.62, 2.30, 3.50, 3.26 and 2.28 times higher than that of the GEL, indicating that the modified films had intermolecular interactions, including but not limited to hydrogen bonding and van der Waals forces [47]. The GP11 had the highest melting and thermal denaturation temperatures, indicating that this ratio of pullulan was the most compatible with gelatin and also had more appropriate hydrogen bonding and intermolecular interactions.

### 3.8. Color and Light Transmission

Color plays a critical role in food packaging, influencing consumer choice [27]. Material light transmission, on the other hand, is closely related to the photosensitivity of the packaging contents. Figure 6 and Table 3 demonstrate the optical properties of different films, including color and optical transmittance. As it can be seen from the table, GEL had the smallest color difference values (ΔE), indicating the best transparency. Thus, the transparency gradually increased with the addition of pullulan, the GP01 being the best transparent among the films containing pullulan. The brightness of the films with multiple gelatin structures was lower; for example, the L* value of the GP31 was only 89.07. The b* value of the GEL was higher than that of the PUL, indicating a more yellowish color, which was determined by the nature of gelatin [30]; therefore, yellow was the base color of all films, and the multiple gelatin structures had a more yellow color, which changed from yellow to blue with the addition of pullulan [27]. In terms of a* value, the film color was redder as the pullulan increased, from −2.16 in GP11 to −1.21 in GP13.

The light transmission properties of the films include visible light transmission rate (VLT), ultraviolet light rejection rate (UVR) and infrared light rejection rate (IRR), with the GEL films having the highest visible light transmission, which was consistent with the color analysis results. The GP11 surprised us by having the lowest VLT and the highest UVR among the blended films, which was good news for oil bagging [3]. The UV barrier of the blended films was more significant than that of the GEL and PUL films, but the color difference results did not indicate that the darker the color, the better the UV and light barrier (GP31), which might be related to the structural morphology of the gelatin and pullulan molecules, as the GP11 had a parallel bonded fibrin structure with less agglomeration and fewer gaps created, thus would have greater light-blocking properties.

### 3.9. Films Antioxidant Activity

The DPPH radical scavenging test is a common method to determine the antioxidant capacity of materials [10]. Figure 7 show the antioxidant activity of GEL, PUL and blended films. The higher the DPPH residual value, the antioxidant activity is weaker. From the figure, the antioxidant activity of GEL was the weakest [22], with a DPPH residual of 97.60% and a scavenging rate of only 2.40% for DPPH radicals, while the scavenging rate of 9.24% for DPPH radicals for PUL, probably due to the presence of phenolic hydroxyl groups in pullulan, resulted in better antioxidant activity (*p* < 0.05), from which it could be concluded that the addition of pullulan would improve the antioxidant properties of gelatin-based edible films. This was demonstrated by the fact that the GP10 without the addition of pullulan had a DPPH residual of 94.33%. In comparison, the GP11 and GP13 had a scavenging rate of 9.92% and 7.21% for DPPH radicals, which were 4.13 and 3.00 times higher than the antioxidant properties of gelatin, respectively, which confirmed the hydroxyl group that appeared to be hydrogen-bonded in the FTIR analysis, precisely because the hydroxyl group formed by pullulan and gelatin molecules had the phenolic hydroxyl-like effect that enhanced the antioxidant properties of the film [48]. This was evident in the GP31, GP13, GP01 and GP11 films with the addition of pullulan.

### 3.10. Protection against Oxidation of Oils and Fats

#### 3.10.1. Changes in Primary Oxidation Products of Liver Oil

Figure 8 show the relevant oxidation indicators during the storage of fish liver oil. Hydroperoxides are the primary oxidation products of oils or fats, and their content is expressed in terms of peroxide values. It can be seen that in the early stage of oxidation (3d), the primary oxidation products were lower in each composite film packaging group than in the blank group (7.82 mmol/kg), probably due to the oxygen barrier. By day 6, the peroxidation values of the GP10 and GP11 groups were significantly lower than those of the other groups. By the middle and late stages (9d and 12d), the GP13 and GP31 groups presented higher peroxidation values, suggesting that an inappropriate pullulan to gelatin ratio could have a pro-oxidation effect [30]. The GP11 group, on the other hand, had a lower peroxide value (7.83 mmol/kg), initially showed a delayed oxidation effect of the oils and fats and had a better effect than the control group, conventional plastic PVC (8.68 mmol/kg) (*p* < 0.05).

#### 3.10.2. Changes in Secondary Oxidation Products of Liver Oil

Hydroperoxides undergo further oxidative decomposition to produce secondary oxidation products. Anisidine is used to detect non-volatile carbonyl compounds such as α and β-unsaturated aldehydes resulting from the oxidation of oils and fats. Malondialdehyde (MDA) is a product of the peroxidation reaction of lipids by the action of oxygen radicals. Both are indicators to characterize the content of secondary oxidation products in oils and fats. As can be seen from Figure 8, the anisidine values (An.V) of 3d for each blended film packaging group were higher than the blank and control, probably because the residual oxygen in the packaging microenvironment of the blended film, after the generation of hydroperoxides, was unable to escape due to the oxygen barrier properties of the films. This residual oxygen continued to remain in the microenvironment for the generation of secondary oxidation products. In contrast, systems where oxygen can diffuse freely were where primary oxidation processes mainly took place, such as the blank group. In the case of MDA, the phenomenon continued until the middle and end of the oxidation. The unsuitable pullulan content of GP13 and GP31 showed a pro-oxidation effect for both primary and secondary oxidation products, while the GP11 group was the better performing group. The GP11 group had the lowest total oxidation value of 41.12, showing good protection against lipids oxidation.

### 3.11. Correlation Analysis between Resistance to Grease Oxidation and Key Properties of Films

We already know that oxygen and UV light are the two main external factors contributing to grease oxidation [1,3], so it is necessary to explore the effect of the film’s shading and oxygen permeability on the four indicators related to grease oxidation. Using Pearson correlation analysis, the results are shown in Figure 9. We found a positive correlation between MDA content and oxygen transmission rate in the early stages of oil bale storage and a negative correlation with UVR, but the effect of OTR was more significant. The better the oxygen permeability, the higher the MDA content in the pre-storage period, which was consistent with the previous experimental results, i.e., the order of oxygen permeability and MDA content on the third day was GP10 > GP31 > GP13 ≈ GP11; meanwhile, the OTR gradually changed to a negative correlation with MDA content as the storage time increased, which explained the higher MDA content in the compounded film packaging group compared to the blank and control groups. The effect of UVR on anisidine was the same as the positive correlation in the early stages, gradually changing to a negative correlation. UVR played a lesser role than OTR in the early stages of oxidation and only started to show its effect in the middle and late stages of oxidation, e.g., by day 9, the higher the UVR, the lower the peroxide value would be. By the late stages of oxidation, UV blocking continued to retard the oxidation of oils and fats.

## 4. Conclusions

In this study, we investigated the film formation mechanism and the effect of resistance to grease oxidation from a practical application. We found that under weak alkaline conditions (provided by sodium tripolyphosphate), the negatively charged gelatin molecules and the polyanionic sodium alginate created space through electrostatic repulsion, providing a place for the cross-linking of pullulan, which would bond with sodium alginate and gelatin molecules through hydrogen bonding. By controlling the ratio of gelatin to pullulan to control the relative strength of hydrogen bonding and electrostatic interactions, we obtained gelatin–sodium alginate–pullulan edible films. A 1:1 ratio of pullulan to gelatin showed the best mechanical strength, oxygen barrier, thermodynamic properties, UV barrier, microstructure observation of parallel bonded fibrous structure and presented the best antioxidant properties. The water solubility also did not differ significantly (*p* > 0.05) from the hydrophilic material. Additionally, it was found that a suitable ratio of pullulan to gelatin (1:1) was able to retard the oxidation of the oil, while the addition of inappropriate pullulan had some pro-oxidation effect. Lastly, the abundant hydrogen bonds in the system can be additionally grafted with antioxidant agents. As a food contact material, the safety and antimicrobial properties of this type of film need to be optimized further, providing alternative options for designing edible inner packaging for convenience foods.

## Figures and Tables

**Figure 1 polymers-14-03199-f001:**
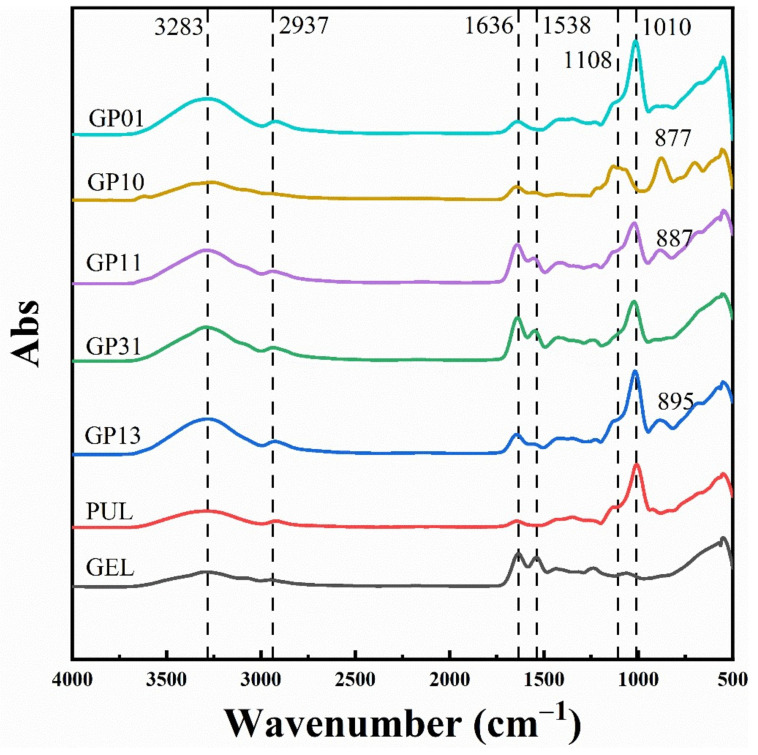
FTIR spectrum of different films.

**Figure 2 polymers-14-03199-f002:**
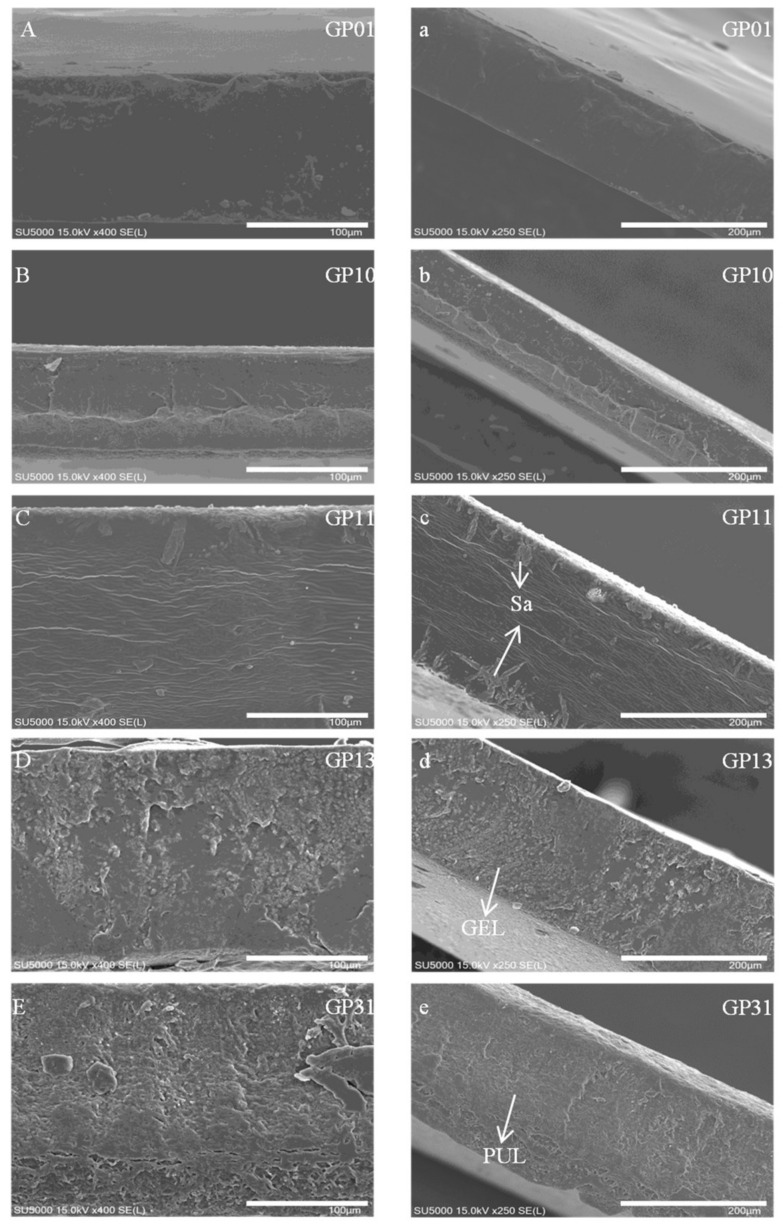
SEM images of the cross-sectional microstructure of different films. (**a**–**e**) are 250× magnification, and (**A**–**E**) are 400× magnification (PUL: Pullulan, GEL: Gelatin, Sa: Sodium alginate).

**Figure 3 polymers-14-03199-f003:**
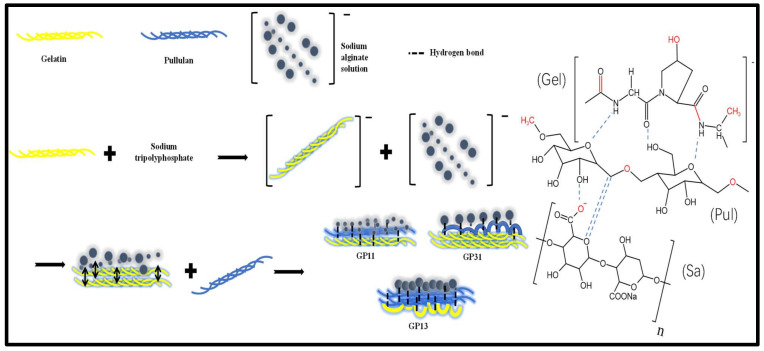
Interaction of pullulan with sodium alginate–gelatin-based films (Schematic diagram of film formation mechanism, Pul: Pullulan, Gel: Gelatin, Sa: Sodium alginate).

**Figure 4 polymers-14-03199-f004:**
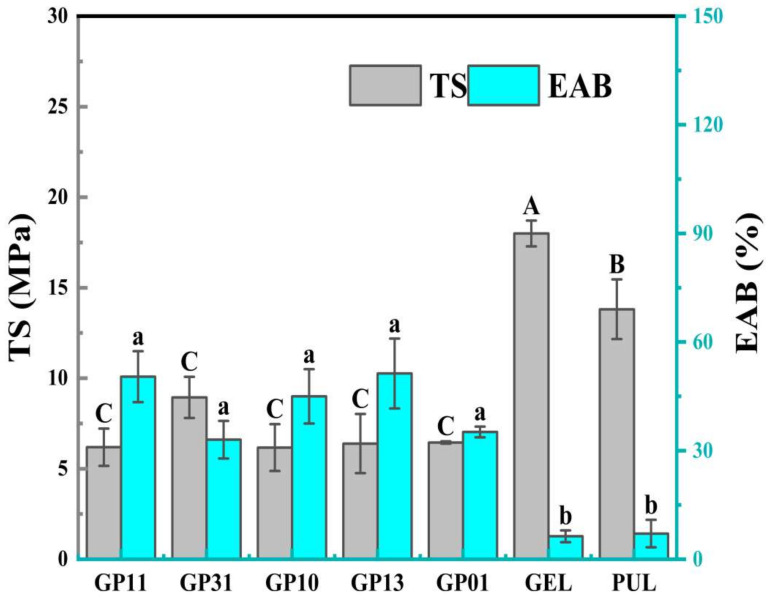
Tensile strength (TS) and elongation at break (EAB) of different films. (Different letters in the same data indicate significant differences (*p* < 0.05).).

**Figure 5 polymers-14-03199-f005:**
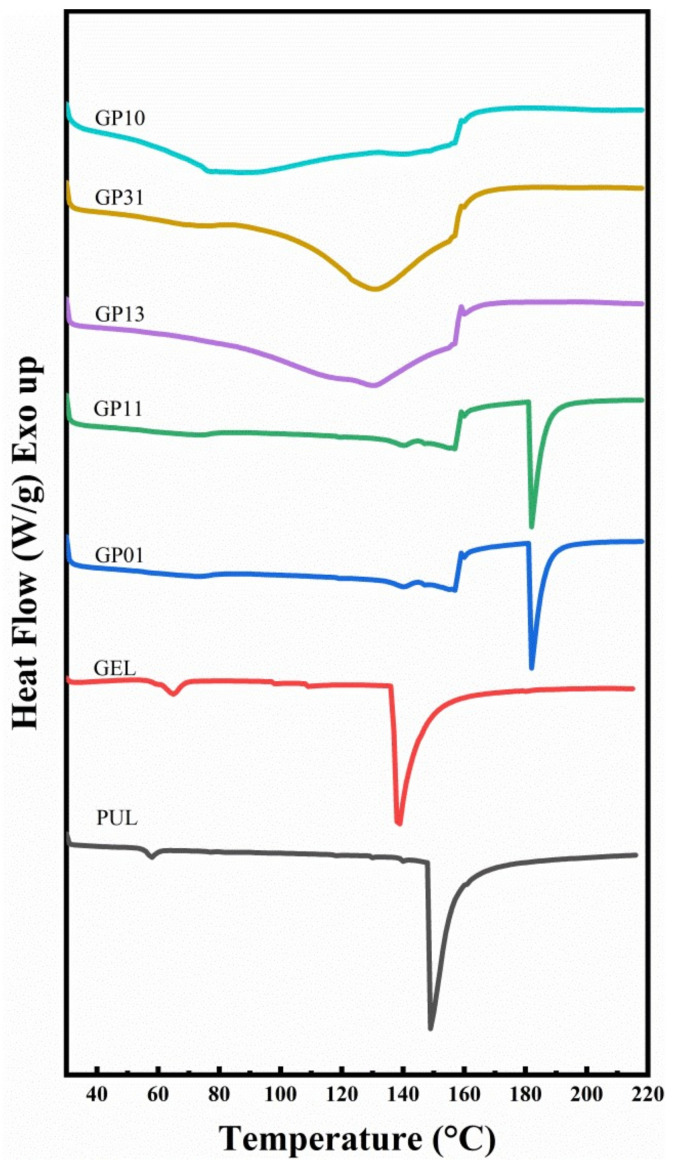
DSC heat flow diagrams for different films.

**Figure 6 polymers-14-03199-f006:**
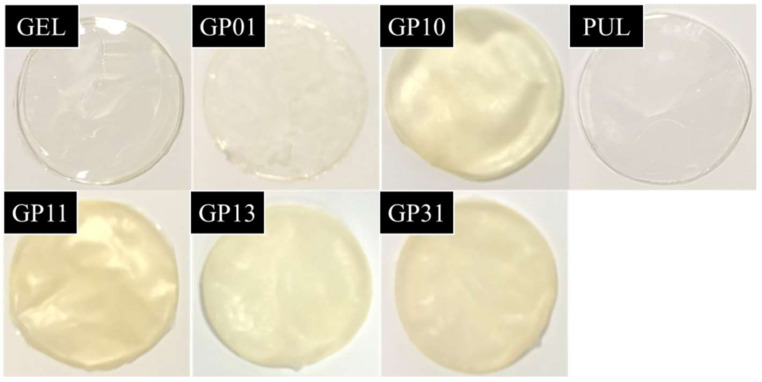
Images of film samples.

**Figure 7 polymers-14-03199-f007:**
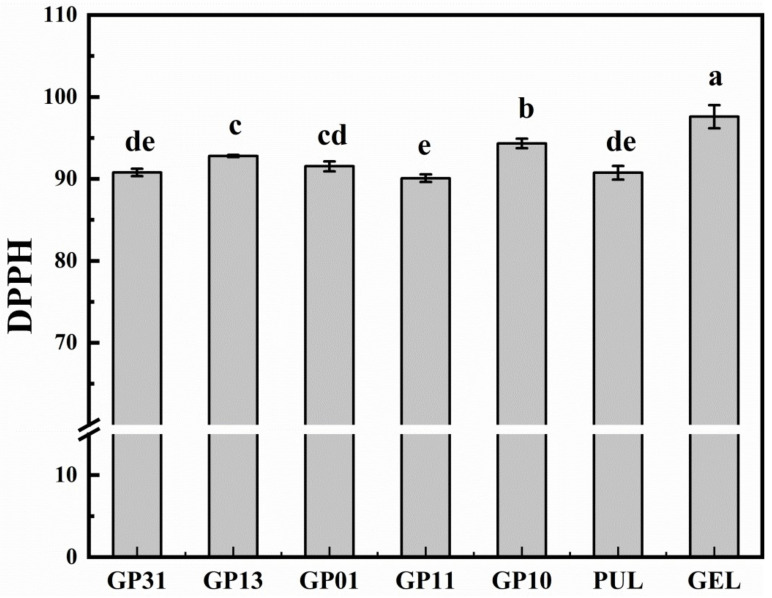
2,2-Diphenyl-1-picrylhydrazyl (DPPH) radical scavenging activity of different films.

**Figure 8 polymers-14-03199-f008:**
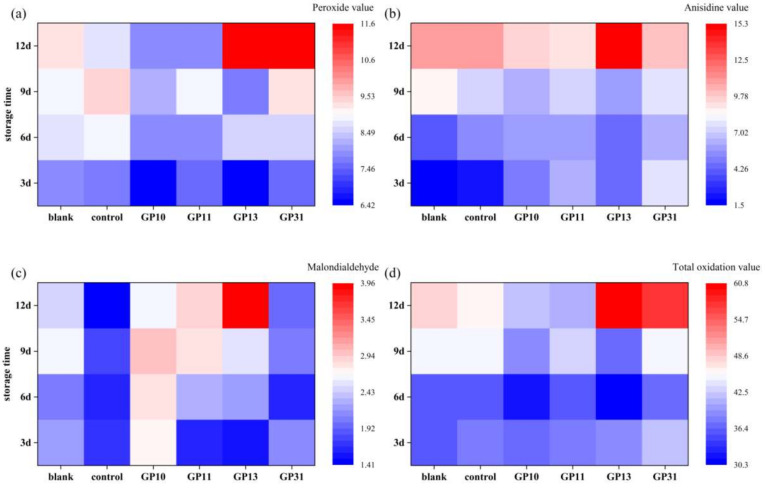
Degree of liver oil oxidation during storage, (**a**) liver oil peroxide value (PV, mmol/kg), (**b**) anisidine value (An.V, dimensionless), (**c**) malondialdehyde (MDA, mg/kg), (**d**) total oxidation value (TOV, dimensionless), at 3d, 6d, 9d, 12d.

**Figure 9 polymers-14-03199-f009:**
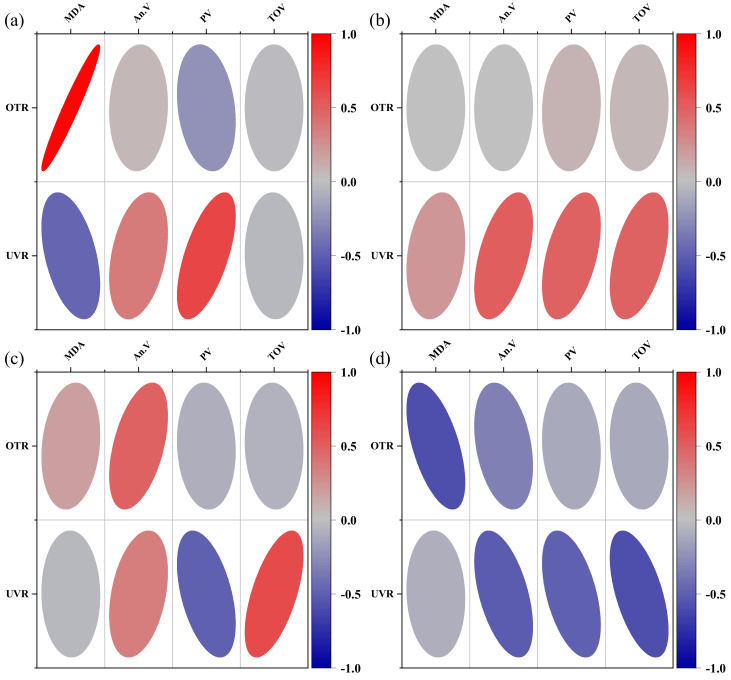
Correlation of grease oxidation indicators with key film properties. (**a**) 3d, (**b**) 6d, (**c**) 9d, (**d**) 12d.

**Table 1 polymers-14-03199-t001:** Films Component.

Component (%)	GP01	GP13	GP11	GP31	GP10	GEL	PUL
Gelatin	0	19.44	38.89	58.33	77.77	100	0
Pullulan	77.77	58.33	38.89	19.44	0	0	100
Sodium alginate	5.56	5.56	5.56	5.56	5.56	0	0
Sodium tripolyphosphate	11.11	11.11	11.11	11.11	11.11	0	0
Glycerol	5.56	5.56	5.56	5.56	5.56	0	0

**Table 2 polymers-14-03199-t002:** Thickness, OTR and WS of various films.

Film Groups	Thickness (mm)	OTR (cm^3^/(m^2^⋅24 h⋅0.1 MPa))	WS (g/s)
GP11	0.201 ± 0.018 ^a^	47.42 ± 13.73 ^ab^	0.10 ± 0.03 ^ab^
GP31	0.179 ± 0.008 ^b^	54.47 ± 8.80 ^a^	0.11 ± 0.01 ^a^
GP10	0.149 ± 0.006 ^c^	57.69 ± 10.31 ^a^	0.07 ± 0.01 ^ab^
GP13	0.145 ± 0.004 ^c^	49.06 ± 5.26 ^ab^	0.10 ± 0.02 ^ab^
GP01	0.138 ± 0.016 ^c^	-	0.07 ± 0.02 ^b^
GEL	0.177 ± 0.009 ^b^	35.86 ± 3.37 ^b^	0.07 ± 0.01 ^b^
PUL	0.143 ± 0.009 ^c^	19.09 ± 3.45 ^c^	0.09 ± 0.01 ^ab^

Abbreviations: Different letters in the same column indicate significant differences (*p* < 0.05). “-” indicates undetectable.

**Table 3 polymers-14-03199-t003:** Color parameters and optical properties of different films.

	*L**	*a**	*b**	Δ*E*	*VLT (%)*	*UVR (%)*	*IRR (%)*
GP11	90.92 ± 0.52 ^ab^	−2.16 ± 0.06 ^c^	11.53 ± 0.56 ^b^	13.03 ± 0.50 ^b^	27.07 ± 4.21 ^c^	94.13 ± 3.50 ^a^	57.67 ± 4.41 ^ab^
GP31	89.07 ± 2.79 ^b^	−2.56 ± 0.41 ^d^	15.33 ± 3.36 ^a^	17.26 ± 4.07 ^a^	42.63 ± 14.68 ^bc^	75.17 ± 27.89 ^a^	69.23 ± 18.08 ^a^
GP10	92.78 ± 0.07 ^a^	−2.09 ± 0.09 ^c^	10.33 ± 0.60 ^b^	11.42 ± 0.60 ^bc^	56.77 ± 4.01 ^b^	72.17 ± 10.96 ^a^	28.93 ± 3.72 ^cd^
GP13	93.26 ± 0.15 ^a^	−1.21 ± 0.10 ^b^	6.45 ± 0.50 ^c^	7.56 ± 0.51 ^d^	66.57 ± 9.94 ^ab^	71.9 ± 11.23 ^a^	20.93 ± 6.79 ^de^
GP01	91.74 ± 1.71 ^ab^	−1.14 ± 0.20 ^b^	6.96 ± 1.16 ^c^	8.53 ± 1.64 ^cd^	45.5 ± 32.09 ^bc^	69.17 ± 23.69 ^a^	44.53 ± 20.74 ^bc^
GEL	93.08 ± 0.19 ^a^	−1.44 ± 0.07 ^b^	7.87 ± 0.47 ^c^	8.96 ± 0.43 ^cd^	90.7 ± 0.95 ^a^	30.6 ± 2.44 ^b^	8.63 ± 0.81 ^e^
PUL	92.8 ± 2.56 ^a^	−0.57 ± 0.10 ^a^	3.87 ± 0.17 ^d^	5.69 ± 1.00 ^d^	91.93 ± 0.06 ^a^	10.57 ± 0.21 ^b^	8.23 ± 0.55 ^e^

Abbreviations: Different letters in the same column indicate significant differences (*p* < 0.05).

## Data Availability

Data presented in this study are available on request from the first author.

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
