# Peer review of "Characterization and Application in Packaging Grease of Gelatin–Sodium Alginate Edible Films Cross-Linked by Pullulan"

_polymers, 2022, doi:10.3390/polym14153199_

Round 1
Reviewer 1 Report
The following few specific comments are suggested to improve the manuscript.
1. The last paragraph of the introduction should be re-written. In this section (lines 82-85), the authors have described the general chemistry between pullulan and sodium tripolyphosphate, and they did not explain what they did in this research. It should be discussed in the last paragraph of the introduction.
2. In lines 144-145, what are g and s?
3. The lines 195-196, 207-208, and 221-222 should be cited by the references.
4. The author state, “ For PUL and GEL membranes, the peaks around 3300 cm-1 and 2937 cm-1 are related to the stretching vibration of –OH.” How does the 2937 cm-1 peak correspond to the stretching vibration of –OH? It should be cited by the references.
5. In table 1, How did the author select these values to make the film? The total of these values in a few columns is over 100.
6. The authors should add pullulan's interaction chemical reaction scheme with sodium alginate-gelatin-based films. What is the role of sodium alginate, sodium tripolyphosphate, and glycerol in this scheme? It should be clearly shown in the scheme. Although the authors have provided a representative interaction in Figure 3, this representative figure is not clearly supporting the FTIR data.
7. In figure 1, the Wavenumber/cm-1 should be written as Wavenumber (cm-1).
8. In figure 5, Exo and endo directions should be mentioned.
9. All images of film samples (transparency and colour) should be added to the manuscript.
Author Response
Response:
Thank you very much for your help to make our manuscript better! We also invited a native English-speaking colleague to check our manuscript.
For more details on responses to comments, please see the attachment.

Reviewer 2 Report
1) Title very long
2) Abstract (DPPH tests) is nt enough FRAP and betacarotene too for comparison 94%
3) using chitosan. [24]. or using chitin [24]. please verify
4) table 1 is not complete when you put - (0 or no activities or what)? plus a lot off repititions
5) 3.2. Microstructure moved to the next page (no title in the botum of the page alone)
6) Table 3 splitted between two pages
with regards
Author Response
Response:
Thank you very much for your help to make our manuscript better!
For more details on responses to comments, please see the attachment.

Reviewer 3 Report
In this paper, the authors investigate the interaction of pullulan with gelatin-sodium alginate-based edible films. This is an ecxellent report-well written, clear and concise. The introduction section provide useful information for the readers. Detailed and systematic experimental studies are presented. The results reported represent a notable advance in the designing of edible inner packaging for convenience foods.
Author Response
Response:
Thank you for your review comments, we are honoured to receive your approval of our manuscript.
This manuscript has been partially revised. Please see the revised version.

Round 2
Reviewer 1 Report
Accept